# Identification and Quantitation Studies of Migrants from BPA Alternative Food-Contact Metal Can Coatings

**DOI:** 10.3390/polym12122846

**Published:** 2020-11-29

**Authors:** Nan Zhang, Joseph B. Scarsella, Thomas G. Hartman

**Affiliations:** Department of Food Science, Rutgers, The State University of New Jersey, 65 Dudley Road, New Brunswick, NJ 08901, USA; joseph.scarsella@rutgers.edu

**Keywords:** bisphenol A, bisphenol C, bisphenol F, tetramethyl bisphenol F, tetramethyl bisphenol F diglycidyl ether, cyclic polyester oligomer, GC-MS

## Abstract

Bisphenol A (BPA)-based epoxy resins have wide applications as food-contact materials such as metal can coatings. However, negative consumer perceptions toward BPA have driven the food packaging industry to develop other alternatives. In this study, four different metal cans and their lids manufactured with different BPA-replacement food-contact coatings are subjected to migration testing in order to identify migratory chemical species from the coatings. Migration tests are conducted using food simulants and conditions of use corresponding to the intended applications and regulatory guidance from the U.S. Food and Drug Administration. Extracts are analyzed by gas chromatography mass spectrometry (GC-MS) and high resolution GC-MS. The migratory compounds identified include short chain cyclic polyester migrants from polyester-based coatings and bisphenol-type migrants including tetramethyl bisphenol F (TMBPF), tetramethyl bisphenol F diglycidyl ether (TMBPF DGE), bisphenol F (BPF), bisphenol C (BPC), and other related monomers or oligomers. The concentration of the migrants is estimated using an internal standard, and validated trimethylsilyl (TMS) derivatization GC-MS methods are developed to specifically quantify TMBPF, BPF, BPC, and BPA in the coatings. The results will aid the safety evaluation of new food-contact material coating technology based on TMBPF chemistry and will provide an important reference for the industry in identifying and quantifying non-BPA coating-borne migrants.

## 1. Introduction

Canned products constitute a major category of the foods and beverages consumed in the United States and abroad. Metal cans are often preferred over other packaging materials such as plastics and glass because metals can offer superior properties. The rigidness and durability of metal cans provide protection for the food or beverage products during manufacturing, shipping, and storage. Metal cans are able to tolerate high temperature and pressure conditions, making them ideal for hot-fill products or retort-sterilized shelf stable products. However, metals may interact with the food or beverage components, resulting in the corrosion of the can containers and deterioration of food quality and safety. Therefore, a polymeric coating is usually applied to the inner food-contact surface to prevent such interaction [1]. Bisphenol A (BPA)-based epoxy resins are one of the most widely used materials for metal can coating [2]. BPA is reacted with epichlorohydrin to form BPA epoxy resins composed of Bisphenol A diglycidyl ether (BADGE) and other epoxy oligomers [3]. These epoxy resins can further polymerize and crosslink, forming 4,4′-methylenediphenol polymer chains that can pack in floccules and provide excellent coating barriers to protect the metals against corrosion [4]. Nevertheless, unreacted BPA, BADGE monomer, and other short-chain epoxy oligomers can remain in the coating products and potentially migrate into the food or beverage. There is extensive literature demonstrating the migration of chemical constituents, including BPA, BADGE, and related oligomers, from epoxy-coated food cans. Analytical methods used to identify and quantify these compounds include HPLC with UV and fluorescence detection [5], UPLC coupled to fluorescence detection and time-of-flight mass spectrometry [6], acylation and silylation derivatization of bisphenols coupled to thermal desorption-GC-MS [7], and HPLC-MS/MS [8,9].

The use of BPA-based coatings in food packaging materials has been greatly affected by negative consumer perceptions since toxicological studies have been published revealing potentially adverse effects of BPA [10,11]. These studies have shown that BPA can act as an endocrine disruptor and increase incidences of health problems such as polycystic ovary syndrome, infertility, precocious puberty, and hormone dependent tumors such as breast and prostate cancer [12,13]. BADGE, as well as BPA, has been demonstrated to be migratory in BPA-based can coating systems; in the 1990s, BADGE was found in several canned food products such as fish in oil and anchovy [14]. BADGE was initially a suspected carcinogen, due to its epoxide moieties which are often highly reactive, demonstrating alkylating properties. Later research, however, showed that BADGE is metabolized to the less toxic BADGE•2H_2_O via enzyme hydrolysis and excreted in both free and conjugated forms [15,16]. Therefore, regulatory limits on BADGE migration are often higher compared to those for BPA. Currently, BPA is banned in many countries for infant feeding bottle applications and epoxy coatings for infant formula packaging, but it is approved for use in many other non-infant specific food contact materials [17,18,19]. In the United States, the Food and Drug Administration (FDA) regulates polymeric and resinous food contact coatings for metal substrates under 21 CFR Section 175.300, in which BPA and BPA-epichlorohydrin epoxy resins are listed as permitted substances [20]. In the European Union, metal can coatings are not specifically regulated, but (EU) 2018/213, which specifically regulates the use of BPA in varnishes and coatings intended to contact with food, lowered the migration limit for BPA from 0.6 to 0.05 mg/kg in food-contact plastics [21]. Based on years of extensive research and hundreds of publications regarding BPA safety, the U.S. FDA assures that the use of BPA in currently approved applications is safe [22]. Nevertheless, consumer concern with BPA in foods and food packaging, as influenced by BPA-free product advertisements, continues to grow.

Chemical manufacturers have developed several novel coatings as replacements for BPA-based coatings. Early experimental formulations were based on other bisphenol compounds such as bisphenol F (BPF) and bisphenol S (BPS). These compounds react with epichlorohydrin analogously to BPA, forming epoxy resins and polymeric coating materials. Similarly to BPA, however, non-reacted monomers from these coatings are highly migratory and have questionable toxicological effects [23,24]. Tetramethyl bisphenol F (TMBPF) is a newly proposed bisphenol-type alternative that appears to be promising. TMBPF reacts in the same manner as other bisphenols, forming similar 4,4-methylenediphenol polymer chain functionality. Several toxicological studies have been published demonstrating very low toxicity of TMBPF [4,25,26].Tetramethyl bisphenol F diglycidyl ether (TMBPF DGE), the reaction product of TMBPF with epichlorohydrin, demonstrated low toxicity in previous in vitro assays as well [25].

Another BPA-free can coating currently being investigated is based on polyester chemistry [27]. Polyester polymer coatings are produced from polycondensation reactions between diol and dicarboxylic acid monomers. Polyesters are commonly used in food packaging materials, and many monomeric raw materials have previously been fully evaluated for toxicity and are compliant with food packaging regulations. Polyester can coatings generally have some resistance to corrosion and good adherence to metal surfaces, but they are more susceptible to hydrolysis than bisphenol epoxy can coatings. Additionally, many byproducts of polyester manufacturing are readily migratory. Low molecular weight compounds such as short chain cyclic polyester oligomers [28,29], polyester hydrolysis products, and manufacturing byproducts of polyester coatings can migrate into food or beverage products, introducing non-intentionally added substances (NIAS). Several studies reporting the migration of polyester monomers and oligomers from polyester can coatings have been published. Analytical methods previously used include HPLC-MS/MS [30,31,32], HPLC with diode array (DAD) or charged aerosol (CAD) detection [30,32], and UPLC-HRAM-MS [31,33].

Apart from toxicity evaluations, suitable migration testing is required to identify and quantify all potential coating-borne migrants in order to assess the risks and safety of BPA replacement can coatings. Migration testing of novel coating materials presents three major challenges. First, non-targeted analyses are required because information regarding coating formulation is often proprietary to coating manufacturers and is unavailable to parties performing analytical testing. In addition, NIAS such as reaction byproducts and degradation products of the coatings may migrate into the food simulant, greatly increasing the complexity of the analyses [34]. Second, the new BPA replacement can coatings may contain novel chemical compounds for which reference standards are commercially available for related quantification analyses. Obtaining reference standards for NIAS is often even more challenging because many compounds can be formed as byproducts of coating manufacture and may not be available as pure standards [35]. Last, even when reference standards are available, establishing validated analytical methods with suitable limits of detection and quantification is considerably difficult and time consuming when extracts contain numerous compounds with a wide range of chemical properties.

In this study, four representative metal can and lid sets manufactured with BPA replacement coatings are analyzed by GC-MS and high resolution accurate mass (HRAM) GC-MS using standardized migration test protocols according to U.S. FDA guidance. The goal of the study was to isolate and identify all potential migrants from the can coatings when they are used in their intended application conditions. Additionally, the study aims to develop validated quantification methods for migrants for which chemical standards are commercially available and compare these validated results with initial semi-quantitative evaluation results. This study presents, to the best of our knowledge, novel HRAM electron impact (EI) mass spectra of tetramethyl bisphenol F and its corresponding diglycidyl ether along with several cyclic polyester oligomers as well as the first reported data regarding the migration of tetramethyl bisphenol F from epoxy can coatings. The study will serve as a reference for analyses of BPA replacement coatings which are based on similar raw materials and chemistries.

## 2. Materials and Methods

### 2.1. Food Packaging Samples

Four can and lid sample sets based on non-BPA formulations were provided by the project sponsors. The chemical composition of the coatings was not known prior to this study. Three of the can and lid sets were made of aluminum and were intended for aqueous non-alcoholic acidic or non-acidic beverage applications which would be hot filled or pasteurized at or below 66 °C (150 °F.) The fourth can and lid set was made of steel and was intended for acidic or nonacidic foods with or without surface fat or oil which would undergo high temperature sterilization at 121 °C (250 °F). All can samples were provided in the final product format before the closure sealing, while all lid samples were provided both in the final product format as well as in flat metal sheet stocks for the ease of sample extraction. The can samples and lids samples were packed separately. It was known that the food can and corresponding lid samples were coated with the same non-BPA-based coating chemistry.

### 2.2. Reagents, Reference Standards, and Materials

Methylene chloride (Optima grade) and glacial acetic acid (ACS certified grade) were purchased from Thermo Fisher Scientific, Waltham, MA, USA. Anthracene-*d_10_*, BPA (≥99%), bisphenol C (BPC) (analytical standard grade), BPF (analytical standard grade) TMBPF (98%), and BSTFA (with 1% trimethylchlorosilane) derivatizing reagent (Sylon BFT) were purchased from Sigma-Aldrich Chemical Co, St. Louis, MO, USA. Ethanol was purchased from Pharmco Products, Brookfield, CT, USA and was re-distilled prior to use. Distilled and de-ionized water used in the study was prepared by double distillation in a glass-lined still followed by activated carbon filtration and de-ionization treatment in a Waters Milli-Q Nanopure system.

### 2.3. Migration Testing of Can and Lidstock Samples

Sections of beverage can wall material and all lid stock samples were cut and sealed into custom stainless steel migration cell assemblies designed according to FDA specifications for food contact material migration testing. The migration cells were comprised of 10 cm × 15 cm stainless steel plates (top and bottom) with a PTFE spacer gasket between the plates. The assembly was bolted firmly together with 12 stainless cap screws around the perimeter. The top plate contained 0.635 cm (0.25 inch) O.D. tube ports for filling and emptying, and the ports were sealed with 0.635 cm (0.25 inch) stainless steel Swagelok caps with PTFE ferrules. The PTFE spacer provided 51 cm^2^ (7.9 in^2^) food contact surface area and 125 mL cavity volume available for extraction with the food simulant solvent. Migration cells were manufactured in-house for this experiment. Samples were sealed into migration cells with the food-contact side facing into the cavity such that only the food-contact surface would have contact with the food simulant. Migration cells were filled with 80 mL of food simulant, giving a volume to surface area ratio of 1.57 mL/cm^2^ (10 mL/in^2^), in accordance with FDA guidelines [36]. For beverage cans, 10% ethanol (ETOH) and 3% acetic acid (AA) were used as food simulants, corresponding to aqueous non-acidic and aqueous acidic food types, respectively. Analytical method blanks consisting of migration cells without samples sealed inside were prepared alongside the samples. Compounds detected in the method blanks were disregarded in the data treatment of test samples. Migration cells were incubated at temperatures corresponding to the appropriate FDA guidance based on the intended filling goods and filling and storage conditions of the test samples. For beverage can and lid stock samples, incubation conditions were 2 h at 66 °C followed by 238 h at 40 °C, in accordance with the FDA condition of use “C” for food packages hot-filled at 66 °C, and stored at room temperature [37].

The non-BPA-coated food can had thick and rigid ribbing on the side walls that prevented sealing into migration cells. For this reason, the food can and lid were extracted simultaneously, and the extract was therefore a composite extract of the can wall and lid. The food can and lid samples were intended for high temperature heat treatment, so they were extracted using a simulated retort procedure in which the food simulant was sealed inside the can, and the sealed cans were incubated in accordance with appropriate FDA guidelines. Food simulants used for the food can samples were 3% AA, 10% ETOH, and 95% ETOH, corresponding to aqueous acidic, aqueous non-acidic, and fatty food types, respectively. The inside surface area and filling volume of the can were measured to be 280 cm^2^ and 350 mL, respectively, giving a volume to surface area ratio of 1.25 mL/cm^2^. Sealed and filled cans were heated in an autoclave to 121 °C and held for 2 h before being transferred to a 40 °C incubator for 238 h, in accordance with the FDA condition of use “A” for high-temperature, retort-processed goods.

Migration cells and sealed cans were removed from incubation and cooled to room temperature. Extracts were drained into 125 mL borosilicate glass media bottles sealed with PTFE-lined closures and processed immediately.

### 2.4. Extract Workup Procedure

For 10% ETOH and 3% AA extracts, 40 mL aliquots of extracts were transferred to 50 mL borosilicate glass test tubes with PTFE-lined closures. Extracts were matrix-spiked with 4.0 µg of anthracene-*d_10_* internal standard (100 parts per billion weight/volume (ppb *w*/*v*)), and 5 mL of methylene chloride was added. The solutions were vigorously extracted and then centrifuged for 30 min at 2500 rpm in order to promote complete phase separation. The methylene chloride fraction was then transferred to a 5.0 mL borosilicate glass conical-bottom vial and evaporated to approximately 400 µL under a gentle stream of nitrogen. The concentrated extracts were analyzed directly by GC-MS.

For 95% ETOH extracts, 10 mL aliquots of extracts were transferred to 50 mL borosilicate glass test tubes with PTFE-lined closures and were diluted with 35 mL of distilled, deionized water. Extracts were matrix-spiked with 10 µg of anthracene-*d*_10_ internal standard (1000 ppb *w*/*v*), and 5 mL of methylene chloride was added. The solutions were vigorously extracted and then centrifuged for 30 min at 2500 rpm in order to promote complete phase separation. The methylene chloride fraction was then transferred to a 5.0 mL borosilicate glass conical-bottom vial and evaporated to approximately 100 µL under a gentle stream of nitrogen. The concentrated extracts were analyzed directly by GC-MS.

### 2.5. GC-MS Analysis Methodology

GC-MS analyses of migration test extracts described above were conducted using a Varian 3400 gas chromatograph directly interfaced to a Thermo-Finnigan TSQ 7000 triple stage quadrupole mass spectrometer or a Thermo 1300 gas chromatograph directly interfaced to a Thermo Scientific GC-Exactive HRAM Orbitrap (Resolution 60,000). Both instruments were equipped with Thermo Xcalibur data system. The Varian GC was equipped with a 30 m × 0.32 mm ZB-5MS capillary column with 0.25 µm film thickness (Phenomenex, Torrance, CA, USA). The Thermo 1300 GC was equipped with a 30 m × 0.32 mm Rxi-5MS capillary column with 0.25 µm film thickness (Restek, Bellefonte, PA, USA). Carrier gas was helium with 10 psi head pressure. The column temperature programs for both GC were 50 °C (3 min) followed by a 10 °C/minute ramp to 320 °C with a total analysis time of 35 min. The injector was maintained at 300 °C, and the heated transfer line was maintained at 320 °C. The GC injector was operated in splitless mode with a 100:1 split ratio septum purge 30 s post-injection. Sample injections were 1.0 µL. The mass spectrometers were operated in positive ion EI ionization mode (70 eV), scanning a mass range of 35–750 once each second.

### 2.6. TMS-Derivatization

The bisphenol compounds identified in the can coatings contain active hydroxyl groups, which cause excessive interaction with the GC column stationary phase, leading to broad, non-symmetrical peak shape and reduced sensitivity. TMS-derivatization was used to generate di-trimethylsilyl (di-TMS) derivatives of the bisphenols in order to prevent these unwanted interactions and improve sensitivity of the method for these compounds. The bisphenol standards and methylene chloride sample extracts were derivatized by adding 100 µL of Sylon BFT reagent and heating for one hour at 85 °C (or 95 °C for TMBPF standard and TMBPF containing sample extracts because the hindered phenol structure of TMBPF requires higher derivatization temperature) in an aluminum heating block with periodic vortexing. The derivatized extracts were cooled to room temperature and re-analyzed by GC-MS.

### 2.7. GC-MS Method Validation

For GC-MS method validation, calibration curves were established for BPA, BPC, BPF, and TMBPF by analyzing TMS-derivatized standards at nine different concentrations in triplicate. Concentrations ranged from 0.1 µg/mL to 100 µg/mL (0.1 ppm *w*/*v* to 100 ppm *w*/*v*), and all standards contained a constant 50 µg/mL anthracene-*d_10_* internal standard. This concentration range was equivalent to 1 ng/mL to 1000 ng/mL (1 ppb *w*/*v* to 1000 ppb *w*/*v*) for the food simulant extracts of a typical metal can sample in the screening analysis as the extracts were concentrated about 100-fold.

The peak area ratios of the target analyte TMS derivatives to internal standard versus target analyte concentration were used for calibration curve plotting and linearity evaluation. Analytical system precision was performed by analyzing 6 replicates of the standard solution at 50 µg/mL (50 ppm *w*/*v*).

Spiking and recovery was performed by matrix-spiking 50 µg of target analyte and 50 µg of anthracene-*d_10_* into 40 mL of 10% ETOH, 40 mL of 3% AA or 10 mL of 95% ETOH prior to the 5 mL methylene chloride back-extraction as described in food simulant work-up procedure. The methylene chloride fraction was then transferred into a 5.0 mL borosilicate glass conical-bottom vial and evaporated to near dryness under a gentle stream of nitrogen. Extracts were then reconstituted with 1 mL of methylene chloride, TMS-derivatized as above, and analyzed by GC-MS.

## 3. Results and Discussion

### 3.1. Identification and Semi-Quantitation of Coating-Borne Migrants

The results of migration studies performed on four commercially manufactured metal cans and their corresponding lids are reported. The identification and semi-quantitation of coating-borne migrants were based on a non-targeted screening analysis approach, where all captured migrant species were analyzed and then quantified via the total ion chromatogram peak area ratio relative to the anthracene-*d_10_* matrix-spiked internal standard assuming a detector response factor of 1.0. Table 1 summarizes the total number of detected migrant peaks from each sample extract along with the average total estimated migratory concentration in micrograms per square centimeter (µg/dm^2^) from replicate analyses. Overall, the beverage can samples showed fewer and lower level migrants compared to other samples.

Twelve coating-borne migrants were identified in this study. Their identification and average estimated migration levels are listed in Table 2. Seven of the migrants were bisphenols or bisphenol-epoxy-related substances and four of them were cyclic polyester oligomers. An additional migrant, triphenyl phosphine oxide (TPPO), was found in the food can and food can lid samples. TPPO is an oxidation product of triphenylphosphine, which is often used as a catalyst in epoxy-phenol polymerization reactions and as a coupling agent that improves coating adhesion to metal surfaces [38,39,40]. TPPO can exist in the final coating product and then migrate into the food system.

TMBPF and TMBPF epoxy migrants were mainly found in the food can and food can lid samples. For the first time, high-resolution GC-MS EI mass spectra of TMBPF and TMBPF DGE migrants were reported (Figure 1). A mass accuracy ±5 ppm and an isotope confidence of <10% difference from the theoretical isotope match (isotope distribution and ratio) were applied on the molecular ions at 256.1460 [C_17_H_20_O_2_]^+^ and 368.1981 [C_23_H_28_O_4_]^+^ to assign the formula (Figure 2). The characteristic electron ionization fragmentation ions at 241.1224 [C_16_H_17_O_2_]^+^, 226.0990 [C_15_H_14_O_2_]^+^, 211.0754 [C_14_H_11_O_2_]^+^, 135.0805 [C_9_H_11_O]^+^, and 91.0543 [C_7_H_7_]^+^ were used for structure elucidation of TMBPF. The characteristic electron ionization fragmentation ions at 353.1746 [C_22_H_25_O_4_]^+^, 335.1640 [C_21_H_22_O_4_]^+^, 293.1173 [C_19_H_16_O_4_]^+^, 191.1065 [C_12_H_15_O_2_]^+^, 175.0753 [C_12_H_15_O]^+^ and 91.0543 [C_7_H_7_]^+^ were used for structure elucidation of TMBPF DGE. The presence of the TMBPF-based coating formulation was also confirmed with the sample supplier after data analysis.

TMBPF can react with epichlorohydrin and form TMBPF DGE and TMBPF monodiglycidyl ether (TMBPF MGE). The epoxy groups of TMBPF DGE can further polymerize with phenols. It can also hydrolyze to form TMBPF DGE•H_2_O and TMBPF DGE•2H_2_O (Figure 3). TMBPF-based epoxy can coating was proposed to be a very promising alternative for BPA. In a previous study, the TMBPF migration level was found below a limit of detection (LOD) at 0.2 ppb (equivalent to 0.03 μg/dm^2^) using 3% AA and 50% ETOH as food simulants under FDA condition of use A, but TMBPF epoxy migrants were not mentioned [4]. While in our study, both TMBPF and TMBPF epoxy migrants were found; they can occur at 0.78 μg/dm^2^ and 5.10 μg/dm^2^, respectively, following extraction under the FDA condition of use A. The higher migration levels and more TMBPF epoxy migrant species can be caused by differences in can coating formulation and application conditions such as the baking temperature and curing time. Several studies also demonstrate that the extent of the bisphenol can coating migration phenomenon may depend on various factors including coating thickness and particle size, time and temperature of retort processing, interactions between the coating and food constituents, and denting or other physical damage to the can [6,41,42].

Other bisphenol migrants including BPA, BPC, and BPF were only found in the beverage can lid samples. Their migration levels were lower compared to the TMBPF migration level in the food can and food can lid sample, likely because the extraction conditions were less extreme. It was notable that the beverage can coatings were not intentionally formulated with BPA. However, it is possible that the final coating product still contains trace amounts of BPA from production line cross-contamination or polymer degradation. They were generally considered as BPA non-intent (BPANI) products.

Short chain cyclic polyester oligomers including cyclic DEG-AA, cyclic DEG-PA, cyclic EG-AA-EG-AA, and cyclic EG-SeA-EG-PA were another group of coating-borne migrant identified in the beverage can and beverage can lid samples. These cyclic polyester oligomers are byproducts of the dicarboxylic acid-diol polycondensation reaction. They do not have free hydroxyl groups to form high molecular weight polymers, and they are low molecular weight, migratory compounds. The most common raw materials in polyester manufacturing include diol monomers, such as diethylene glycol (DEG) and ethylene glycol (EG), and dicarboxylic acid monomers, such as adipic acid (AA), sebacic acid (SeA) and phthalic acid (PA). Therefore, the short chain cyclic polyester oligomer migrants based on these diols and dicarboxylic acids are frequently observed in polyester-based coatings or lamination adhesive products [24]. Short chain cyclic polyesters are considered as NIAS, and chemical standards are usually not commercially available for related method validation and safety evaluation studies.

### 3.2. Quantitation of Bisphenol Monomer Migrants

In this study, TMS derivatization GC-MS analytical methods were developed for BPA, BPC, BPF, and TMBPF quantitation. High-purity standards were available for these compounds. For BPA, BPC, and BPF, previous studies utilized both GC-MS and LC-MS for quantitation analysis. GC-MS analytical methods for TMBPF have not been reported previously. The developed method was validated for linearity, limit of detection (LOD), limit of quantitation (LOQ), recovery, and precision (Table 3).

Calibration curves were obtained by plotting the peak area ratio of BPA-di-TMS/IS, BPC-di-TMS/IS, BPF-di-TMS, or TMBPF-di-TMS/IS versus the concentrations of standard solutions of BPA, BPC, BPF, or TMBPF, respectively. The dynamic range of calibration was 1.00–100.00 µg/mL (1.00–100.00 ppm *w*/*v*) and the coefficient of determination R^2^ equaled 0.9913, 0.9957, and 0.9904, respectively, for BPA, BPC, and BPF. The dynamic range of TMBPF calibration was 0.500–100.00 µg/mL (0.500–100.00 ppm *w*/*v*) and the coefficient of determination R^2^ equaled 0.9915.

The LOD determined with a signal-to-noise (S/N) of 3 for BPA-di-TMS, BPC-di-TMS, and BPF-di-TMS were 0.501 µg/mL, 0.499 µg/mL, and 0.501 µg/mL, respectively, where the noise was selected from peaks adjacent to the peak of the target analytes. This concentration was equivalent to approximately 5 ng/mL (5 ppb *w*/*v*) for the food simulant extracts of a typical metal can sample in the screening analysis as the extracts were concentrated about 100-fold. The LOQ determined with a S/N of 10 for BPA-di-TMS, BPC-di-TMS and BPF-di-TMS were 1.001 µg/mL, 0. 998 µg/mL and 1.002 µg/mL, respectively. The LOD for TMBPF-di-TMS (S/N = 3) was 0.100 µg/mL, which is equivalent to approximately 1 ng/mL (1.0 ppb *w*/*v*) for the food simulant extracts of a typical metal can sample in the screening analysis. The LOQ for TMBPF-di-TMS (S/N = 10) was 0.500 µg/mL.

Analytical system precision was accessed by running six replicate analyses of a solution containing target analytes at 50 µg/mL (50 ppm *w*/*v*). The precision of BPA, BPC, BPF, and TMBPF had an RSD of 9.01%, 10.24%, 8.67% and 9.63%, respectively. The mean back-fit to calibration for BPA, BPC, BPF, and TMBPF was 53.88 ± 4.85 µg/mL, 50.47 ± 5.17 µg/mL, 54.69 ± 4.74 µg/mL, and 52.34 ± 5.04 µg/mL, respectively.

Spiking and recovery of BPA, BPC, BPF, and TMBPF was performed in triplicate in 10% ethanol and 3% acetic acid food simulants at a 1.25 µg/mL concentration level and in 95% ethanol food simulant at a 5.00 µg/mL concentration level. The recovery results of BPA spiked in 10% ethanol, 95% ethanol, and 3% acetic acid were 96.12% ± 8.35%, 99.69% ± 8.71%, and 76.31% ± 5.92%, respectively. The recovery results of BPC spiked in 10% ethanol, 95% ethanol, and 3% acetic acid were 81.45% ± 7.42%, 96.64% ± 6.16%, and 94.21% ± 7.99%, respectively. The recovery results of BPF spiked in 10% ethanol, 95% ethanol, and 3% acetic acid were 74.60% ± 5.45%, 93.38% ± 8.07%, and 86.05% ± 9.41%, respectively. The recovery results of TMBPF spiked in 10% ethanol, 95% ethanol, and 3% acetic acid were 122.78% ± 9.38%, 103.99% ± 9.02%, and 77.41% ± 4.42%, respectively.

The validated method was applied to re-analyze the can and lid samples which showed bisphenol-type migrants. The resulting BPA, BPC, and BPF concentrations in the beverage can lid samples were all below the limit of detection at 5.0 ppb *w*/*v* (0.785 µg/dm^2^). The resulting TMBPF concentrations in the food can 10% ethanol, 3% acetic acid, and 95% ethanol extracts are all between LOD at 1.0 ppb *w*/*v* (0.125 µg/dm^2^) to LOQ at 5.0 ppb *w*/*v* (0.625 µg/dm^2^). The concentrations of TMBPF in the re-analyzed 10% ethanol and 3% acetic acid food can lid sample extracts were below 0.157 µg/dm^2^, while the TMBPF concentration in the re-analyzed 95% ethanol food can lid sample extract was between LOD at 1 ppb *w*/*v* (0.157 µg/dm^2^) to LOQ at 5.0 ppb *w*/*v* (0.785 µg/dm^2^).

## 4. Conclusions

In this study, 12 coating-borne migrants based on bisphenol or polyester chemistry were found from non-BPA food-contact metal can coatings. Four of the migrants were based on TMBPF, a proposed low toxicity BPA replacement bisphenol monomer. They were identified and reported for the first time in food can packaging intended for high temperature application. Validated TMS derivatization GC-MS methods were developed to quantify the concentrations of bisphenol monomer migrants including BPA, BPC, BPF, and TMBPF from food-contact metal can coatings.

The identification data of coating-borne migrants including the molecular structure, estimated migration levels, and high resolution EI mass spectra can be used as references for the food industry. The quantitation analysis of BPA, BPC, BPF, and TMBPF provided a validated GC-MS method able to determine the concentrations of bisphenol-type migrants from food packaging, and can assist manufacturers in performing systematic quality control of their BPA alternative can coating products.

## Figures and Tables

**Figure 1 polymers-12-02846-f001:**
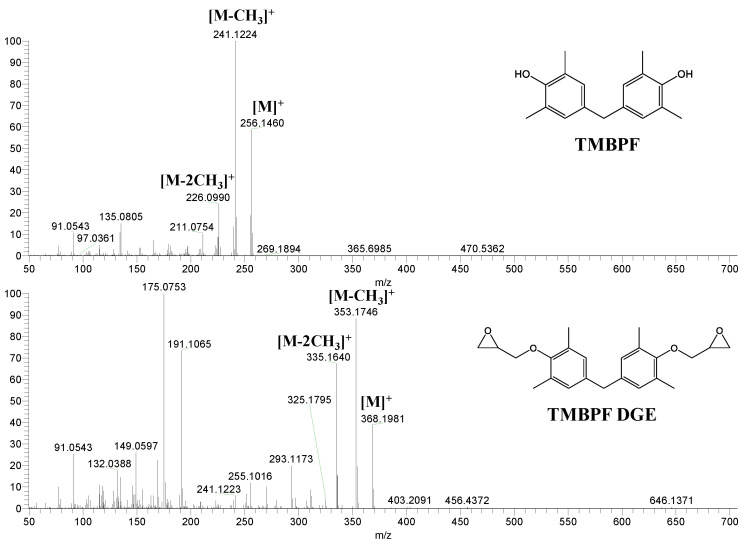
High resolution GC-MS Spectra for tetramethyl bisphenol F (TMBPF) and tetramethyl bisphenol F diglycidyl ether (TMBPF DGE).

**Figure 2 polymers-12-02846-f002:**
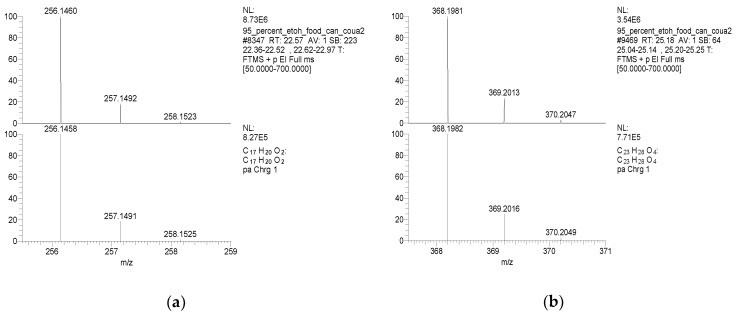
Isotopic pattern comparisons of molecular ions (Top) and theoretical molecular ions (Bottom). (**a**) TMBPF; (**b**) TMBPF DGE.

**Figure 3 polymers-12-02846-f003:**
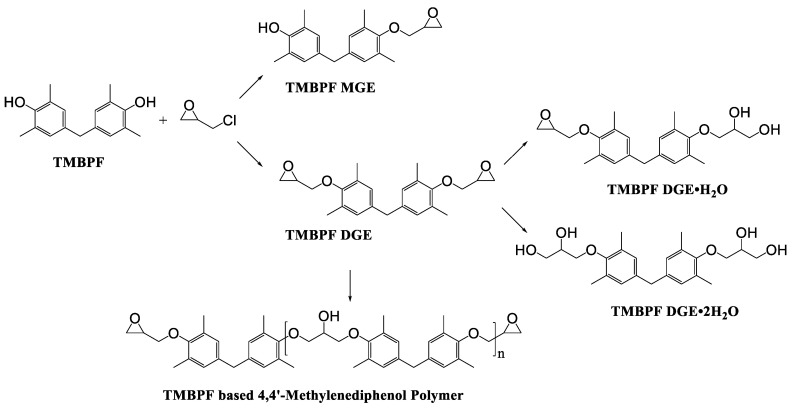
Epoxy TMBPF Formation, polymerization and hydrolysis.

**Table 1 polymers-12-02846-t001:** Number of detected migrants and estimated concentration of total migrants from metal cans.

Samples	Number of Detected Migrants			Estimated Level of Total Migrants (μg/dm^2^)		
	3% AA	10% ETOH	95% ETOH	3% AA	10% ETOH	95% ETOH
Food can set	38	42	41	34.13	51.65	140.03
Food can lid alone	26	25	26	3.37	3.16	41.42
Beverage can 1	4	2	-	0.44	0.08	-
Beverage can 2	7	2	-	0.80	0.08	-
Beverage can 3	6	6	-	0.58	0.23	-
Beverage can lid 1	26	12	-	42.73	50.02	-
Beverage can lid 2	15	10	-	1.36	3.81	-
Beverage can lid 3	15	8	-	1.49	0.47	-

**Table 2 polymers-12-02846-t002:** Food-contact coating-borne migrants from metal cans (name, molecular formula, structure, mass, five major EI mass spectra fragment ion peak percent abundance, estimated concentration, and source).

Name ^1^	Molecular Formula and Structure	Mass	5 Major EI Fragments (% Abundance)	Estimated Level (μg/dm^2^)	Source
TMBPF	C_17_H_20_O_2_ 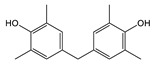	256	241(100.00) 256(59.39) 226(24.63) 255(18.67) 242(17.99)	0.04–0.78	Food can setFood can lid alone
TMBPF MGE	C_20_H_24_O_3_ 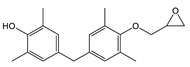	312	312(100.00) 239(45.30) 254(41.26) 209(30.63) 135(27.34)	0.03–1.33	Food can lid alone
TMBPF DGE	C_23_H_28_O_4_ 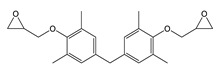	368	175(100.00) 353(86.00) 191(66.51) 335(63.68) 368(35.86)	0.04–5.10	Food can setFood can lid alone
TMBPF DGE•H_2_O	C_23_H_30_O_5_ 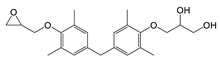	386	241(100.00) 256(58.60) 135(28.19) 75(19.47) 386(16.59)	0.04–0.40	Food can setFood can lid alone
TPPO	C_18_H_15_OP 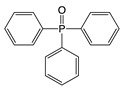	278	277(100.00) 278(58.99) 77(27.03) 201(25.00) 199(15.50)	0.05–2.91	Food can setFood can lid alone
BPA	C_15_H_16_O_2_ 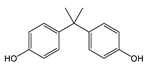	228	213(100.00) 228(24.99) 119(20.01) 214(17.26) 91(10.02)	0.04–0.06	Beverage can lid 1 and lid 3
BPC	C_17_H_20_O_2_ 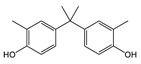	256	241(100.00) 133(22.57) 256(19.37) 242(18.21) 77(8.76)	0.01–0.03	Beverage can lid 1 and lid 3
BPF	C_13_H_12_O_2_ 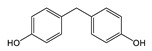	200	200(100.00) 104(77.47) 199(46.37) 183(21.16) 94(14.20)	0.04	Beverage can lid 1
Cyclic DEG-AA	C_10_H_16_O_5_ 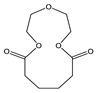	216	173(100.00) 55(78.00) 99(35.86) 84(32.54) 56(25.04)	0.02–0.04	Beverage Can 1,2 and 3
Cyclic DEG-PA	C_12_H_12_O_5_ 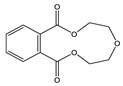	236	149(100.00) 193(98.57) 104(33.37) 76(24.76) 148(17.20)	0.10	Beverage Can 2
Cyclic EG-AA-EG-AA	C_16_H_24_O_8_ 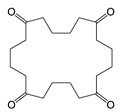	344	173(100.00) 99(61.02) 55(41.78) 113(32.68) 111(24.45)	0.05	Beverage can lid 1
Cyclic EG-SeA-EG-PA	C_22_H_28_O_8_ 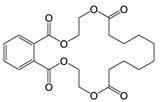	420	237(100.00) 149(81.43) 193(64.50) 104(62.39) 148(37.35)	5.05–9.77	Beverage can lid 1

^1^ TMBPF, tetramethyl bisphenol F; TMBPF MGE, tetramethyl bisphenol F monoglycidyl ether; TMBPF DGE, tetramethyl bisphenol F diglycidyl ether; TMBPF DGE•H2O, monohydrolysed tetramethyl bisphenol F diglycidyl ether, BPA, bisphenol A; BPC, bisphenol C; BPF, bisphenol F; TPPO, triphenyl phosphine oxide; EG, ethylene glycol; DEG, diethylene glycol; SeA, sebacic acid; PA, phthalic acid; IPA.

**Table 3 polymers-12-02846-t003:** Analytical performance of developed method for bisphenol monomer migrants.

Target Analyte	Linearity (R^2^)	LOD (µg/mL)	LOQ (µg/mL)	Precision (%)	Spiking and Recovery (Recovery%, n = 3)		
					10% ETOH	95% ETOH	3% AA
BPA	0.9913	0.501	1.001	9.01	96.12 ± 8.35	99.69 ± 8.71	76.31 ± 5.92
BPC	0.9957	0.499	0.998	10.24	81.45 ± 7.42	96.64 ± 6.16	94.21 ± 7.99
BPF	0.9904	0.501	1.002	8.67	74.60 ± 5.45	93.38 ± 8.07	86.05 ± 9.41
TMBPF	0.9915	0.100	0.500	9.63	122.78 ± 9.38	103.99 ± 9.02	77.41 ± 4.42

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
