# Peer review of "Identification and Quantitation Studies of Migrants from BPA Alternative Food-Contact Metal Can Coatings"

_polymers, 2020, doi:10.3390/polym12122846_

Round 1

Reviewer 1 Report

The article submitted for review is of certain interesting and can provide useful information for both the scientific and industrial community.

Moreover, the paper is well written, well organized and addresses an interesting subject.

Some remarks:

A deep review is necessary to complete the manuscript : the authors can compare the obtained results with those reported in the literature.

A detailed analysis of the recent literature data can improve the scientific value of the manuscript.

Author Response

Dear Reviewer,

Thank you very much for your comments and suggestions. We modified our manuscript and included more references in our introduction section and results and discussion section.
Please let us know if you have any other questions or suggestions.
Thanks again.

Nan Zhang, Joseph B. Scarsella and Thomas G. Hartman

Reviewer 2 Report

Different migrants from BPA-based coatings are identified and quantified in this contribution. The applied analytical techniques are adequate and their application is well explained and motivated. The work is fully empirical but well organized. Even though the novelty content is quite limited, the results are original enough. Such results can be of interest to producers and regulatory entities, probably much less to the polymer science community. The manuscript is carefully and clearly written. Overall, I evaluate this contribution as publishable.

Two minor remarks.
- page 2, line 49: ...shown that BPA can (act) as an endocrine disruptor...
- page 5, line 238: the detector response factor is assumed equal to 1. Is this reasonable? Can Authors provide some support about the reliability of this assumption?

Author Response

Dear Reviewer,

Thank you very much for your comments. We revised corrected our manuscript based on your remark 1.

For your second remark, the response factor assumption only applies to our screening analyses where standards were not available for the identified migrants and Anthracene-d10 is a commonly used internal standard in migration studies. (Examples: [1]. Fasano, E., Bono-Blay, F., Cirillo, T., Montuori, P., & Lacorte, S. (2012). Migration of phthalates, alkylphenols, bisphenol A and di (2-ethylhexyl) adipate from food packaging. Food Control27(1), 132-138; [2]. Nam, S. H., Seo, Y. M., & Kim, M. G. (2010). Bisphenol A migration from polycarbonate baby bottle with repeated use. Chemosphere79(9), 949-952; [3]. Scarsella, J. B., Zhang, N., & Hartman, T. G. (2019). Identification and migration studies of photolytic decomposition products of UV-photoinitiators in food packaging. Molecules24(19), 3592.)

Please let us know if you have any other questions or suggestions.

Thanks again.

Nan Zhang, Joseph B. Scarsella and Thomas G. Hartman

Reviewer 3 Report

The work of Zhang et al. examined an important topic. The experiments of the authors are well described.  The results were discussed and the conclusions were also sound. I recommend publication of this work in Polymers after a minor revision.

[1] The comparison of the present results with previous investigations on literature would increase the novelty of this study.

[2] Relevant literature should be cited. The following examples are highly recommended;

[a] K. Hamad, M. Kaseem, M. Ayyoob, J. Joo, F. Deri, Polylactic acid blends: The future of green, light and tough, Progress in polymer Science.  85 (2018) 83-127.

[b] J. Bott, A. Stormer, P. Albers, Investigation into the release of nanomaterials from can coatings into food, Food Packaging and Shelf Life 16 (2018) 112–121.

[c] S. Noureddine El Moussawi, R. Ouaini, J. Matta, H. Chébib, M. Cladière, V. Camel, Simultaneous migration of bisphenol compounds and trace metals in canned vegetable food, Food Chem. 288 (2019) 228–238.

Author Response

Dear Reviewer,

Thank you very much for your comments and suggestions. We modified our manuscript and included more references as comparison. Your suggested references are very helpful in our results and discussion section.

Please let us know if you have any other questions or suggestions.

Thanks again.

Nan Zhang, Joseph B. Scarsella and Thomas G. Hartman